# Cardiomyocyte Apoptosis Is Associated with Contractile Dysfunction in Stem Cell Model of *MYH7* E848G Hypertrophic Cardiomyopathy

**DOI:** 10.3390/ijms24054909

**Published:** 2023-03-03

**Authors:** Alexander M. Loiben, Wei-Ming Chien, Clayton E. Friedman, Leslie S.-L. Chao, Gerhard Weber, Alex Goldstein, Nathan J. Sniadecki, Charles E. Murry, Kai-Chun Yang

**Affiliations:** 1Institute for Stem Cell and Regenerative Medicine, School of Medicine, University of Washington, Seattle, WA 98109, USA; 2Center for Cardiovascular Biology, University of Washington, Seattle, WA 98109, USA; 3Department of Medicine/Cardiology, University of Washington, Seattle, WA 98109, USA; 4Cardiology/Hospital Specialty Medicine, VA Puget Sound HCS, Seattle, WA 98108, USA; 5Department of Mechanical Engineering, University of Washington, Seattle, WA 98195, USA; 6Department of Lab Medicine and Pathology, University of Washington, Seattle, WA 98195, USA; 7Department of Bioengineering, University of Washington, Seattle, WA 98195, USA

**Keywords:** hypertrophic cardiomyopathy, dilated cardiomyopathy, MYH7, p53, engineered heart tissue, apoptosis, contractile dysfunction, induced pluripotent stem cells

## Abstract

Missense mutations in myosin heavy chain 7 (*MYH7*) are a common cause of hypertrophic cardiomyopathy (HCM), but the molecular mechanisms underlying *MYH7*-based HCM remain unclear. In this work, we generated cardiomyocytes derived from isogenic human induced pluripotent stem cells to model the heterozygous pathogenic *MYH7* missense variant, E848G, which is associated with left ventricular hypertrophy and adult-onset systolic dysfunction. *MYH7*^E848G/+^ increased cardiomyocyte size and reduced the maximum twitch forces of engineered heart tissue, consistent with the systolic dysfunction in *MYH7*^E848G/+^ HCM patients. Interestingly, *MYH7*^E848G/+^ cardiomyocytes more frequently underwent apoptosis that was associated with increased p53 activity relative to controls. However, genetic ablation of *TP53* did not rescue cardiomyocyte survival or restore engineered heart tissue twitch force, indicating *MYH7*^E848G/+^ cardiomyocyte apoptosis and contractile dysfunction are p53-independent. Overall, our findings suggest that cardiomyocyte apoptosis is associated with the *MYH7*^E848G/+^ HCM phenotype in vitro and that future efforts to target p53-independent cell death pathways may be beneficial for the treatment of HCM patients with systolic dysfunction.

## 1. Introduction

Hypertrophic cardiomyopathy (HCM), characterized by unexplained left ventricular hypertrophy, affects 1 in 500 individuals in the general population [1,2] While the left ventricular function is generally preserved or hyperdynamic, it is increasingly acknowledged that the primary defect in some cases is impaired contractile function [1,2,3,4,5,6]. Mutations in myosin heavy chain 7 (*MYH7*), encoding a sarcomeric thick filament protein, are common genetic causes for HCM, accounting for 33% of the cases [7,8]. Recently, mavacamten, a myosin inhibitor, was found to improve symptoms in HCM patients with preserved to hyperdynamic systolic function and left ventricular outflow tract obstruction by reducing contractility; however, mavacamten is contraindicated in patients with reduced ejection fraction as it can further worsen systolic function [9]. Because ~10% of HCM patients develop systolic dysfunction, this class of medication is not an option for them [9,10]. In order to develop novel therapies for patients with HCM and systolic dysfunction, a better understanding of the molecular mechanisms that govern this disease is needed.

Cardiomyocyte apoptosis has been observed in various models of cardiac diseases [11,12,13,14,15,16]. The activation of tumor suppressor p53, a major driver of intrinsic apoptosis, has been implicated in the progression of cardiac hypertrophy at both the cellular and tissue level [17,18,19,20,21,22,23,24]. In a human induced pluripotent stem cell-derived cardiomyocyte (hiPSC-CM) model of the *MYH7*^R403Q/+^ HCM associated with hypercontractile function, p53 inhibition partially rescued cardiomyocyte survival but did not normalize the hypercontractile function in cardiac microtissues [17]. Given that the role of p53 in HCM associated with hypocontractile function is unknown, we hypothesize that inhibition of p53 in this setting will improve cardiomyocyte survival and overall contractile function.

In a previous work, patients harboring the heterozygous *MYH7^E848G/+^* variant presented with adult-onset familial systolic dysfunction and mild ventricular wall thickening [6]. Since that publication, an additional family member presented with significant left ventricular hypertrophy that met criteria for HCM. Given that HCM is now established in this family, the rest of the *MYH7*^E848G/+^ family members exhibiting at least 1.3 cm wall thickening now meet diagnostic criteria for HCM as per the 2020 American College of Cardiology and American Heart Association HCM Guidelines [25]. Thus, *MYH7*^E848G/+^ hiPSC-CM is an ideal model for testing the role of p53 in HCM associated with hypocontractile function. Here, we improve upon the prior viral transgenesis approach using clustered regularly interspaced short palindromic repeat (CRISPR)/Cas9 editing of patient-derived hiPSCs to generate isogenic lines expressing the pathogenic *MYH7* E848G variant fused with enhanced green fluorescent protein (EGFP) to better understand the pathophysiology of *MYH7*^E848G/+^-based HCM associated with hypocontractile function. This model recapitulated the clinical phenotype as we observed increased cardiomyocyte hypertrophy and decreased tissue contractility in both patient-derived and isogenic hiPSC-CMs expressing *MYH7*^E848G/+^. In cardiomyocytes derived from the hiPSCs, we found that the *MYH7* E848G variant increased cytotoxicity, apoptosis markers, and p53 expression, but genetic ablation of *TP53* did not restore contractile function or cardiomyocyte survival. Overall, our findings suggest that in HCM patients with systolic dysfunction, cardiomyocyte apoptosis contributes to impaired tissue contractility with p53-independent cell death as a potential mechanism.

## 2. Results

### 2.1. Generation of Isogenic βMHC-EGFP-Expressing hiPSC-CMs Using CRISPR/Cas9 Editing

Since the publication of our last study, another family member in the original study presented (patient Id) with clear left ventricular (LV) septal wall thickening on echo (1.9 cm) and severe LV systolic dysfunction (ejection fraction 39%) at age 57 [6] (Figure 1A). Based on the diagnostic criteria for HCM as recommended, because one family member has clear HCM phenotype, the rest of the family members with at least 1.3 cm wall thickening would now meet diagnostic criteria for HCM (patient Ia) [25]. To study the effects of the *MYH7* E848G variant in the context of isogenic gene-edited hiPSCs in vitro, we leveraged previously generated human induced pluripotent stem cells (hiPSCs) derived from an HCM patient (HCM IIb) and a non-variant family member (WT Ib) (Figure 1A). To generate isogenic hiPSC lines with fluorescent tracking of beta-myosin heavy chain (βMHC), the protein encoded by *MYH7*, we designed a gene editing strategy to create hiPSC lines expressing βMHC-EGFP fusion proteins (Figure 1B and Appendix A). 

Enrichment with the *Pgk*-puromycin cassette improved the gene-editing efficiency such that ~10% of the colonies screened were correct (Appendix A). By knocking *MYH7* cDNA in-frame with the sequence of EGFP into the endogenous *MYH7* locus of HCM IIb *MYH7*^E848G/+^-hiPSCs, we enabled direct native control of the expression and tracking of the βMHC-EGFP fusion protein (Figure 1B). With this approach, we generated four isogenic βMHC-EGFP-expressing hiPSC lines with all combinations of WT and E848G homozygous and heterozygous alleles, with one allele EGFP-tagged: *MYH7*^WT/WT-EGFP^, *MYH7*^WT/E848G-EGFP^, *MYH7*^E848G/WT-EGFP^, and *MYH7*^E848G/E848G-EGFP^ (Figure 1C). Notably, our editing approach yielded successfully edited clones as verified by Sanger sequencing with high efficiency, with cumulatively 10 of 76 picked clones (13.2%) across the four lines (Appendix A) correctly gene-edited. Green striated sarcomeres were visible in confocal microscopy in each line upon successful differentiation into hiPSC-CMs (Figure 1C). Western blot confirmed the presence of two βMHC protein bands of roughly equal intensity, corresponding with untagged and EGFP-tagged βMHC (Appendix A), suggesting no preferential expression of one allele over the other. Differentiation of *MYH7*^WT/WT-EGFP^ and *MYH7*^WT/E848G-EGFP^ hiPSCs generated hiPSC-CMs of roughly 80% purity at differentiation day 25 as measured with EGFP^+^ fraction via flow cytometry (Appendix A), with no difference in relative expression of EGFP-tagged βMHC as measured using EGFP^+^ mean fluorescent intensity (Appendix A). The establishment of these hiPSC-CM lines enabled various lines of inquiry related to the effects of *MYH7*^E848G/+^ on cardiomyocyte behavior in vitro, while also providing evidence of the utility of our approach for creating multiple edits to study a *MYH7* variant.

### 2.2. MYH7 E848G Variant Increases Cardiomyocyte Size and Reduces Engineered Heart Tissue Contractility 

First, we sought to address whether our *MYH7*-EGFP lines recapitulated in vitro the key measures of hypertrophy and hypocontractility present in vivo. *MYH7*^WT/WT-EGFP^ and *MYH7*^WT/E848G-EGFP^ hiPSC-CMs were seeded for 7 days in monolayer on Matrigel. Immunocytochemistry revealed significantly increased *MYH7^WT/E848G-EGFP^* two-dimensional projected area (3100 ± 170 μm^2^) relative to *MYH7^WT/WT-EGFP^* projected area (2870 ± 110 μm^2^), an 8.0% increase (Figure 1D,E). This finding is consistent with a previously reported increase of 10% in cell size in *MYH7*^R403Q/+^ hiPSC-CMs [17]. All three isogenic E848G-expressing lines exhibited increased forward scatter area (11.1 ± 4.6%, 9.4 ± 5.8%, 13.4 ± 3.7% increase) relative to *MYH7^WT/WT-EGFP^* as measured with flow cytometry (Figure 1F,G), confirming the immunocytochemistry findings. This matches the cellular hypertrophy observed in patient-derived hiPSC-CMs, as HCM IIb cardiomyocytes also had an increased forward scatter area (13.2 ± 6.9% increase) relative to WT Ib cardiomyocytes (Figure 1H,I). To assess tissue contractility, we used the K3 configuration of three-dimensional engineered heart tissues (EHTs) on flexible polydimethylsiloxane (PDMS) microposts (Appendix A), which approximates moderate afterload as previously described [26]. The maximum twitch force at week 3 of EHT casting was significantly weaker in all three isogenic E848G-expressing lines (147.0 ± 19.5 μN, 173.0 ± 8.4 μN, 161.7 ± 18.0 μN) relative to *MYH7*^WT/WT-EGFP^ (238.2 ± 17.4 μN) (Figure 1J). With patient-derived hiPSC-CMs, the maximum twitch force at week 3 of EHT casting was also significantly weaker in HCM IIb EHTs (118.9 ± 23.7 μN) relative to WT Ib EHTs (287.4 ± 27.7 μN), mirroring the findings in EHTs with the isogenic lines (Figure 1K). In sum, our isogenic and patient-derived *MYH7* E848G lines demonstrated both cardiomyocyte hypertrophy and tissue hypocontractility, thus serving as a useful in vitro model of *MYH7*^E848G/+^ HCM.

### 2.3. MYH7 E848G Variant Reduces hiPSC-CM Survival in Monolayer Culture

To expedite the maturation process and, thus, the expression and effects of *MYH7* E848G variant, we tested two different cardiomyocyte culture media: one with an RPMI base and low calcium concentration (0.4 mM Ca^2+^), and one with a DMEM base and more approximately physiological calcium concentration (1.8 mM Ca^2+^). After 10 days of treatment, EGFP intensity as measured with flow cytometry was significantly increased for *MYH7*^WT/WT-EGFP^ hiPSC-CMs cultured in DMEM-based media (93.6 ± 3.2% increase) relative to those in RPMI-based media (Appendix A); FSC area also increased with DMEM-based media (13.9 ± 8.0% increase) relative to RPMI-based media (Appendix A), suggesting increased maturation with the DMEM-based high calcium media. Moving forward, we used this DMEM-based media for all monolayer experiments.

While culturing the *MYH7*^E848G/+^ cardiomyocytes, we noted a significant loss of the mutant cardiomyocytes during prolonged culture (Figure 2A). WT Ib and HCM IIb patient lines and *MYH7-EGFP* isogenic lines were monolayer-cultured for two weeks in DMEM-based media. Intriguingly, lines without *MYH7* E848G had a negligible difference in cTnT^+^ or EGFP^+^ total cell count over time, but those with *MYH7* E848G had a significant reduction in cTnT^+^ or EGFP^+^ total cell count (Figure 2B). To further explore the manner of cell death, *MYH7*^WT/WT-EGFP^ and *MYH7*^WT/E848G-EGFP^ hiPSC-CMs were stained with terminal deoxynucleotidyl transferase dUTP nick end labeling (TUNEL). The fraction of TUNEL^+^ nuclei in EGFP^+^ cells after 3 days of DMEM-based media treatment was significantly higher in *MYH7*^WT/E848G-EGFP^ (10.2 ± 0.8%) relative to *MYH7*^WT/WT-EGFP^ hiPSC-CMs (3.7 ± 0.3%), suggesting apoptosis was the cause of cell death (Figure 2C,D). Combined, these data indicate *MYH7*^E848G/+^ reduces hiPSC-CM viability when cultured on a stiff tissue culture surface, suggesting that the *MYH7*^E848G/+^ cardiomyocytes may be susceptible to increased afterload.

### 2.4. MYH7^*E848G/+*^ Reduces hiPSC-CM Survival and Increases Cardiomyocyte Size in EHTs

We posited that the reduced monolayer viability of hiPSC-CMs expressing the *MYH7* E848G variant may translate to the EHT environment and help explain the impaired hypocontractility phenotype. Thus, we utilized a previously described method for papain-based digestion of EHTs into single cells for analysis with fluorescence-activated cell sorting (FACS) (Figure 2E) [27]. There was a significant decrease in the EGFP^+^ fraction of the sorted EHT population in *MYH7*^WT/E848G-EGFP^ tissues at both 1 week (13.2 ± 6.0%) and 3 weeks (7.6 ± 0.7%) post-cast, relative to the EGFP^+^ fraction in *MYH7*^WT/WT-EGFP^ tissues (27.4 ± 0.9%, 20.9 ± 1.6%) as detected with flow cytometry (Figure 2F,G). Although cardiomyocyte loss was persistent over time in the EHTs, there was differential survival in the 1st week of culture that was not seen between 1 and 3 weeks of culture in EHTs, indicating that once the EHTs had compacted to steady state, the *MYH7* E848G expressing cardiomyocytes no longer exhibited increased cytotoxicity compared to the control line. This is in contrast to cardiomyocytes plated on stiff tissue culture plastic, where we found a persistent genotype-dependent decrease in survival over time (Figure 2B), further suggesting that the genotype-dependent cardiomyocyte apoptosis is, in part, due to increased afterload. We next examined if the cardiomyocytes exhibited a hypertrophic response in the 3D environment. *MYH7*^WT/E848G-EGFP^ EHTs yielded EGFP^+^ cardiomyocytes with increased forward scatter area (8.4 ± 1.7% increase) relative to those sorted from *MYH7*^WT/WT-EGFP^ EHTs (Figure 2H), indicating that an E848G-induced hypertrophic response was also present in the 3D environment.

### 2.5. p53 and Associated Markers of Intrinsic Apoptosis Are Elevated in MYH7^*WT/E848G-EGFP*^ hiPSC-CMs

Given the TUNEL results and the reduced survival of *MYH7*^WT/E848G-EGFP^ hiPSC-CMs in both monolayer and EHT environments, we decided to further explore apoptotic signaling. As such, we obtained protein lysates from *MYH7*^WT/WT-EGFP^ and *MYH7*^WT/E848G-EGFP^ hiPSC-CMs cultured with 5 days of DMEM-based media and probed them on a human apoptosis antibody array for 43 protein targets (Figure 3A). We saw a significant upregulation of p53 (90.0 ± 8.5% increase) and associated downstream signaling elements Bad (65.2 ± 14.4% increase), Bax (73.8 ± 25.0% increase), Caspase-3 (83.6 ± 24.5%), Caspase-8 (54.1 ± 6.4%), and p21 (92.9 ± 27.3%) in *MYH7*^WT/E848G-EGFP^ hiPSC-CMs relative to *MYH7*^WT/WT-EGFP^ hiPSC-CMs, suggesting that p53 pathway activity and apoptotic markers are elevated in hiPSC-CMs with the *MYH7* E848G variant (Figure 3B and Appendix A, Table 1). p53 protein expression remained elevated (43.2 ± 11.0% increase) in *MYH7*^WT/E848G-EGFP^ hiPSC-CMs after 10 days of DMEM-based media treatment, corroborating the results of the apoptosis antibody array (Figure 3C,D). Notably, we did not see significant differences in TNFα, TNFβ, or FasL, ruling out extrinsic apoptosis as a mechanism for reduced viability (Table 1). At the transcriptional level, *TP53* expression was elevated after 10 days of DMEM-based media treatment in *MYH7*^WT/E848G-EGFP^ hiPSC-CMs (78.1 ± 9.3% increase) relative to *MYH7*^WT/WT-EGFP^, suggesting p53 upregulation is due to increased transcriptional activity (Figure 3E). In EHTs, FACS-sorted EGFP^+^ cells from *MYH7*^WT/E848G-EGFP^ tissues had higher *TP53* expression (67.0 ± 29.0% increase) relative to those from *MYH7*^WT/WT-EGFP^ tissues (Figure 3F). In sum, *MYH7*^E848G/+^ hiPSC-CMs in both 2D and 3D contexts have elevated p53 signaling activity that is associated with increased cardiomyocyte apoptosis.

### 2.6. TP53 Ablation Does Not Rescue Contractile Function, Cardiomyocyte Survival, or Cellular Hypertrophy in EHTs with MYH7 E848G-Expressing hiPSC-CMs

We sought to determine whether p53 activity was necessary for the observed effects of *MYH7* E848G on contractility and cardiomyocyte survival in our *MYH7-EGFP* hiPSC-CM EHT model. Thus, we used CRISPR/Cas9 gene editing to knock out *TP53* in *MYH7*^WT/E848G-EGFP^ hiPSCs, generating a new *MYH7*^WT/E848G-EGFP^ *TP53*^−/−^ hiPSC line (Figure 4A). Three single guide RNAs (sgRNAs) targeting Exon 5 of *TP53* were used to ablate both *TP53* alleles as evidenced by Sanger sequencing, generating two clonal *TP53*^−/−^ hiPSC lines (Figure 4B and Appendix A). Notably, *MYH7*^WT/E848G-EGFP^ *TP53^−/−^* EHTs (146.4 ± 21.6 μN) had no significant difference in maximum contractile force relative to *TP53^+/+^* EHTs at cast week 3 (147.0 ± 19.5 μN), indicating that p53 activity is not driving impaired contractility in our model (Figure 4C). When these EHTs were sorted by FACS, EGFP^+^ percentage at cast week 1 (11.5 ± 4.5%, n = 4 casts) and total EGFP^+^ counts (47,100 ± 8300 EGFP^+^ cells) in *TP53^−/−^* EHTs were similar to those in *TP53^+/+^* EHTs (13.2 ± 6.0%, 49,700 ± 14,600 EGFP^+^ cells), indicating cardiomyocyte survival was unaffected by ablating p53 activity (Figure 4D,E). Cast week 3 *TP53^−/−^* EHTs (5.1 ± 1.9%, 22,800 ± 4600 EGFP^+^ cells) remained indistinguishable from corresponding *TP53^+/+^* EHTs (7.6 ± 0.7%, 26,000 ± 6100 EGFP^+^ cells). The increased forward scatter area of EGFP^+^ cardiomyocytes in *TP53^+/+^* EHTs (8.4 ± 1.8% increase) persisted in *TP53^−/−^* EHTs (13.6 ± 5.3% increase), indicating that *TP53* ablation does not rescue cellular hypertrophy (Figure 4F). Thus, p53 activity does not appear to be necessary for reduced contractility, cardiomyocyte cytotoxicity, or cellular hypertrophy in our *MYH7* E848G HCM EHT model, and specific targeting of p53 does not restore the healthy phenotype.

## 3. Discussion

In this work, we generated isogenic hiPSCs with *MYH7*-EGFP fusion expression in the endogenous *MYH7* locus, with or without the *MYH7* E848G variant. Our editing approach provides a couple advantages. First, the use of patient-derived hiPSCs with *MYH7*^E848G/+^ as the parental cell line ensured that the corrected and variant isogenic lines have the same patient-derived genetic background. Approaches which use previously established wild-type lines as the base cell line do not capture the same genetic background as a patient-derived model. Second, this gene-editing strategy leverages an antibiotic enrichment cassette that significantly reduces the number of colonies needed to be screened, thereby permitting the generation of multiple isogenic hiPSC lines with *MYH7* variants.

In our isogenic hiPSC-CM model, the *MYH7* E848G variant increased cell size, reduced cardiomyocyte survival, and reduced tissue contractility in three-dimensional culture. These findings correlated with reduced survival, cellular hypertrophy, and impaired tissue contractility in our patient-derived non-fluorescent hiPSC-CM lines. The cellular hypertrophy and decrease in cardiomyocyte survival has been reported in an hiPSC-CM model of the *MYH7*^R403Q/+^ HCM associated with hypercontractile function [17]. In that study, p53 activity was elevated, and inhibition with the small molecule pifithrin partially rescued cardiomyocyte survival, but it did not normalize contractile function. We also observed increased p53 activity in our hypocontractile HCM model, and we genetically ablated *TP53* to interrogate the role of p53. To our knowledge, this is the first study to fully ablate *TP53* expression in the context of HCM-associated cytotoxicity and impaired tissue contractility. We believe genetic ablation provides a definitive answer on the role of p53 in *MYH7*^E848G/+^ HCM associated with systolic dysfunction compared to alternative methods that utilize small molecules or viral transgenesis [17,21]. We have demonstrated that reduced cardiomyocyte survival and tissue hypocontractility are independent of p53 activity in our *MYH7* E848G model of HCM with hypocontractile function. This does not rule out p53’s role in other HCM-causative *MYH7* variants or other sarcomeric variants.

This work represents the first attempt to leverage EHT dissociation [27] to interrogate hiPSC-CM survival, hypertrophy, and expression at a cellular level in the context of an HCM-causative variant with hypocontractile function cultured in a 3D cardiac organoid. Notably, *MYH7* E848G increased cytotoxicity and cell size in the three-dimensional context, demonstrating that the variant effects in two-dimensional culture are also present in a more relevant, 3D environment.

In sum, we have shown that the *MYH7* E848G HCM-causative variant associated with hypocontractile function yields cardiomyocyte hypertrophy with reduced survival and tissue contractility in a p53-independent manner, suggesting that future efforts to target p53-independent apoptotic mechanisms may be beneficial for the treatment of HCM associated with hypocontractile function.

## 4. Materials and Methods

### 4.1. Monolayer Culture of hiPSCs

hiPSCs were cultured in mTeSR+ (STEMCELL Technologies, 100-0276, Cambridge, MA, USA) supplemented with 50 U/mL penicillin/streptomycin (Invitrogen, 15140122, Waltham, MA, USA) on plates coated with 80 μg/mL Matrigel (Corning, 356231, Lot 1242001, Corning, NY, USA) at 5% CO_2_ and 37 °C. Cells were fed every other day and passaged with 500 μM EDTA (Invitrogen, 15575-038, Waltham, MA, USA) before differentiation was morphologically evident. Media were supplemented with 10 μM ROCK inhibitor (SelleckChem, Y27632, Houston, TX, USA) for first 24 h post-passage.

### 4.2. CRISPR/Cas9 Editing of Patient-Derived hiPSCs

Patient-derived hiPSCs corresponding to the non-variant *MYH7* (WT Ib) and heterozygous *MYH7* E848G variant were previously generated [6]. A total of 1000 k pelleted hiPSCs were mixed with 1 μL 10 μM SP-dCas9-VPR (Addgene, 63798, Watertown, MA, USA), 9 μL Buffer R2 (STEMCELL Technologies, 100-0691, Cambridge, MA, USA), 1 μL 30 μM of gRNA (Table 2), and 1.5 μg of pJet-MYH7-EGFP-PGK-PuroR plasmid (Supplementary File S1). For *TP53* ablation, pJet plasmid was omitted. Cells were electroporated at 1400 V for 20 ms with a 10 μL tip using the Neon Transfection System (Thermo Fisher, MPK5000, Waltham, MA, USA). Transfected cells were plated in mTeSR+ without penicillin/streptomycin, and supplemented with CloneR2 (STEMCELL Technologies, 100-0691, Cambridge, MA, USA) and 10 μM ROCK inhibitor on plates coated with 80 μg/mL Matrigel. Cells were fed every other day with mTeSR+ with 0.175 μg/mL puromycin dihydrochloride (Thermo Fisher, A1113803, Waltham, MA, USA) and replated at 88 cells/cm^2^ in a 10 cm plate coated with 80 μg/mL Matrigel for colony picking. Clones were replated in 96-well plates for expansion and genomic DNA harvesting.

### 4.3. Generation of hiPSC-CMs

hiPSCs were seeded at 65k/cm^2^ in mTeSR+ with 10 μM ROCK inhibitor on plates coated with 80 μg/mL Matrigel (Day-2). After 48 h (Day 0), media were replaced with RBA media (RPMI with L-glutamine (Invitrogen, 11875-119, Waltham, MA, USA), 500 μg/mL bovine serum albumin (BSA; Sigma, A9418-50G, St. Louis, MO, USA), 213 μg/mL ascorbic acid (Sigma, A8960-5G)) supplemented with 5 μM Chiron 99021 (Cayman Chemical, 13122, Ann Arbor, MI, USA). After 48 h (Day 2), media were replaced with RBA media supplemented with 2 μM Wnt C59 (SelleckChem, S7037, Houston, TX, USA). After 48 h (Day 4), media were replaced with unsupplemented RBA media. After 48 h (Day 6), media were replaced with RPMI-based cardiomyocyte media (RPMI with L-glutamine, B27 supplement with insulin (Invitrogen, 175044, Lot 2181371, Waltham, MA, USA), 50 U/mL penicillin/streptomycin) and replaced every other day until Day 20. hiPSC-CMs were dissociated with 0.5% trypsin (Invitrogen, 15090046, Waltham, MA, USA) in 500 μM EDTA with 25 μU DNAse I (Sigma, 260913-25MU, St. Louis, MO, USA) and replated at 65k/cm^2^ in RPMI-based cardiomyocyte media with 5% FBS on 20 μg/mL Matrigel-coated plates. hiPSC-CMs were metabolically enriched for 5 d with daily feeding with DMEM without glucose or L-glutamine (Invitrogen, F530S, Waltham, MA, USA), and supplemented with 4 mM Sodium L-lactate (Sigma, 71718-10G, St. Louis, MA, USA). hiPSC-CMs were frozen at Day 25 in Cryostor CS10 (Sigma, C2874-100mL, St. Louis, MA, USA) at −80 °C or immediately used for EHTs.

### 4.4. Monolayer Culture of hiPSC-CMs

Day 25 hiPSC-CMs were thawed at 500 k/cm^2^ in RPMI-based cardiomyocyte media with 5% FBS on 10 μg/mL Matrigel-coated plates. Cells were fed with RPMI-based cardiomyocyte media on Day 26 and 28. On Day 30, hiPSC-CMs were dissociated with 0.5% trypsin in 500 μM EDTA with 25 μU DNAse I and replated at 250 k/cm^2^ in RPMI-based cardiomyocyte media with 5% FBS on 10 μg/mL Matrigel-coated plates. After 24 h (Day 31) and every ensuing 48 h, cells were fed with DMEM-based cardiomyocyte media (DMEM with high glucose (Invitrogen, 10313021, Waltham, MA, USA), B27 supplement with insulin, 50 U/mL penicillin/streptomycin).

### 4.5. Casting of Engineered Heart Tissues (EHTs)

EHTs were cast on polydimethylsiloxane (PDMS) microposts as previously described [26,27] Briefly, Sylgard 184 Elastomer Base and Curing Agent (Dow, 1317318, Midland, MI, USA) were mixed at a 10:1 ratio and cured in a custom 3D-printed mold for 18 h at 65 °C, with one flexible post and one glass rod filled stiff post per set of posts, 6 posts per array. Cured post arrays were removed from the mold and trimmed of excess PDMS. A total of 500k Day 25 hiPSC-CMs and 100k human Hs27a stromal cells were mixed with 3 U/mL thrombin from bovine plasma (Sigma, T4648, St. Louis, MO, USA) and 5 mg/mL bovine fibrinogen (Sigma, E8630, St. Louis, MO, USA) in 100 μL EHT media (sterile filtered RPMI, B27 supplement, 5 g/L aminocaproic acid (Sigma, A2 504-256-100G, St. Louis, MO, USA), penicillin/streptomycin). The cell slurry was added into 2% agarose wells between posts in a 24-well plate and incubated for 80 min at 37 °C, 5% CO_2_. Then, 350 μL EHT media was added to the wells, and tissues were incubated for 10 min at 37 °C, 5% CO_2_. Posts were carefully moved to a fresh 24-well plate in 2 mL EHT media, and tissues were cultured on posts for 3 weeks with media change every other day.

### 4.6. Analysis of EHT Contractile Force

Videos with 5 s duration of paced EHTs were analyzed as previously described [26,27]. To do so, 24-well metal electrode trays with 2 mL Tyrode solution (1.8 mM calcium chloride (Sigma, C4901, St. Louis, MO, USA), 1.0 mM magnesium chloride (Sigma, 1374248, St. Louis, MO, USA), 5.4 mM potassium chloride (Fisher, P330-500, Hampton, NH, USA), 140 mM sodium chloride (Sigma, S5886-1KG, St. Louis, MO, USA), 0.33 mM monobasic sodium phosphate (Fisher, P284-500, Hampton, NH, USA), 10 mM HEPES (Invitrogen, 15630-080, Waltham, MA, USA), 5 mM dextrose (Sigma, D9434, St. Louis, MO, USA) in H_2_O) per well were incubated for 30 min at 37 °C, 5% CO_2_. EHT post arrays were transferred to the electrodes and paced for 5 s with 1.5 Hz, 10 V, 20 ms pulses with 45 fps videos captured using live brightfield microscopy. Maximal twitch force was calculated based on peak length displacement of tissues using previously published MATLAB code.

### 4.7. PCR Amplification and Sequencing

Genomic DNA was isolated from hiPSC subclones using a DNeasy Blood and Tissue Kit (Qiagen, 69506, Hilden, Germany). A *MYH7* fragment containing mutation was amplified through PCR using Q5 High-Fidelity DNA Polymerase (New England Biolabs, M0491L, Ipswich, MA, USA) and 500 nM forward and reverse primers (Table 3). PCR products were run on 1% agarose gels and extracted using a Fermentas Gel Extraction Kit (Invitrogen, K0692, Waltham, MA, USA). Sanger sequencing was performed by Eurofins Genomics (Louisville, KY, USA).

### 4.8. Immunocytochemistry

hiPSC-CMs were seeded at 25 k/cm^2^ in Matrigel-coated 4-well chamber slides (Millipore, PEZGS0416, Burlington, MA, USA) and cultured for 72 h in DMEM-based cardiomyocyte culture media. Cells were fixed with 4% paraformaldehyde for 5 min at room temperature and permeabilized with 0.2% Triton X-100 (Sigma, X100-100ML, St. Louis, MO, USA) in 1× phosphate buffer saline (PBS) for 5 min at room temperature. Cells were rinsed twice with 1× PBS and incubated for 10 min in the dark at room temperature with 1:2000 Hoechst 33342 (Thermo Fisher, H3570, Waltham, MA, USA). Cells were imaged with 40× objective using a custom Nikon ECLIPSE Ti spinning disk confocal microscope with a Yokogawa W1 spinning disk head (Yokogawa, CSU-W1, Tokyo, Japan), using 405 and 488 nm lasers. Images were captured using NIS Elements AR software (Version 5.02.01, Nikon, Tokyo, Japan).

### 4.9. Western Blot

#### 4.9.1. Lysate Preparation

Monolayer-cultured hiPSC-CMs were rinsed twice with 1× PBS and lysed with Pierce RIPA lysis buffer (Thermo Fisher, 89901, Waltham, MA, USA) supplemented with Halt protease and phosphatase inhibitor (Invitrogen, 78443, Waltham, MA, USA) and dithiothreitol (Roche, 3483-12-3, Basel, Switzerland). Lysates were rocked at 4 °C for 20 min and centrifuged 10 min at 15,000× *g*, with supernatant collected. Total protein concentration of supernatant was assessed through a Bradford assay (Bio Rad, 5000006, Hercules, CA, USA) with 560 nm absorbance per manufacturer’s protocol using BSA standards (Thermo Fisher, 23208, Waltham, MA, USA). Lysates were diluted with 4× Laemmli SDS Sample Buffer (Bio Rad, 1610747, Hercules, CA, USA) to 1 μg/μL total protein concentration.

#### 4.9.2. Electrophoresis and Staining

Lysates were loaded at 15 μg in Mini Protean 4–15% polyacrylamide gels (Bio Rad, 4508084, Hercules, CA, USA). Electrophoresis was run at 120 V for 50 min in 1× tris-glycine-SDS running buffer (25 mM tris base (Sigma, T1503-1KG, St. Louis, MO, USA), 190 mM glycine (Fisher, BP381-1, Hampton, NH, USA), 0.1% sodium dodecyl sulfate (Fisher, BP243-1, Hampton, NH, USA)). Proteins were transferred to Immobilon-P membranes (Millipore, IPVH85R, Burlington, MA, USA) at 120 V for 65 min at 4C in 1× tris-glycine-methanol transfer buffer (25 mM tris base, 190 mM glycine, 20% methanol (Fisher, A412P-4)). Membranes were blocked with 5% milk (Bio Rad, 1706404, Hercules, CA, USA) in 1× tris buffer saline with Tween-20 (TBST; 20 mM tris base, 150 mM Tween-20 (Sigma, P9416-100 mL, St. Louis, MO, USA)) for 120 min at RT. Membranes were washed 3× with 1× PBS and incubated with appropriate primary antibody (Table 4) diluted in 4% BSA (Sigma, A9418-50G) in 1× PBS overnight at 4 °C. Membranes were washed 3× with 1× PBS and incubated with appropriate secondary antibody diluted in 1% BSA in 1× TBST for 60 min at RT. Membranes were washed 3× with 1× PBS and incubated with Clarity Max ECL substrate (Bio Rad, 1705061, Hercules, CA, USA) for 5 min. Chemiluminescence images were obtained using the ChemiDoc imaging system (Bio Rad, 17001401, Hercules, CA, USA). Volumetric band intensities were analyzed using Image Lab software (Version 6.1.0, Bio Rad, Hercules, CA, USA).

### 4.10. TUNEL Staining and Analysis

hiPSC-CMs were seeded at 25k/cm^2^ in Matrigel-coated 4-well chamber slides (Millipore, PEZGS0416, Burlington, MA, USA) and cultured for 72 h in DMEM-based cardiomyocyte culture media. Cells were fixed with 4% paraformaldehyde and stained using the Click-IT Plus TUNEL Assay kit (Invitrogen, C10619, Waltham, MA, USA) with Alexa Fluor 647 secondary per manufacturer’s protocol. Cells were rinsed 2× with 1× PBS and incubated for 10 min in the dark at room temperature with 1:2000 Hoechst 33342. Cells were rinsed twice with 1× PBS and imaged at 40× with spinning disk confocal microscopy as above with 405, 488, and 640 nm lasers. Images were segmented using ImageJ, and Hoechst+/TUNEL+ nuclei in EGFP+ cells were quantified.

### 4.11. Flow Cytometry and FACS

#### 4.11.1. Flow Cytometry

Cells were trypsinized, centrifuged, and resuspended as above. For flow assays, cells were fixed with 4% paraformaldehyde for 5 min. For staining of patient line hiPSC-CMs with cardiac troponin T (cTnT), cells were incubated with cardiac troponin T APC-conjugated antibody (Miltenyi Biotec, 130-120-403, Bergisch Gladbach, Germany) or REA control human IgG1 APC-conjugated isotype antibody (Miltenyi Biotec, 130-120-709, Bergisch Gladbach, Germany) for 1 h at room temperature in the dark in 0.75% saponin (Sigma, 558255-100G, St. Louis, MO, USA) and 5% FBS in 1× PBS. Cells were rinsed with 1× PBS, centrifuged for 3 min at 300× *g*, and resuspended in 5% FBS in 1× PBS for analysis using a FACSCanto cytometer (BD Biosciences, San Jose, CA, USA). Populations were serially gated for FSC-A/SSC-A, FSC-H/FSC-W, and GFP+/AmCyan- to identify EGFP-expressing iPSC-CMs.

#### 4.11.2. FACS of EHTs

At cast week 1 or week 3, EHTs were rinsed with 1× PBS and carefully removed from PDMS posts using forceps. Tissues were placed in 1 mL of papain-based dissociation solution (40 U/mL papain from C. papaya (Sigma, 76220-25G, St. Louis, MO, USA), 5.5 mM L-cysteine HCl monohydrate (Sigma, C7880-500MG, St. Louis, MO, USA), 1 mM EDTA (Fisher, 02-002-790), 0.5% beta-mercaptoethanol (Sigma, M6250, St. Louis, MO, USA), 1× PBS) (28). Tissues in dissociation solution were incubated for 10 min at 37 °C, 5% CO_2_ and gently triturated into single cells. Dissociation was halted with 5% FBS in RPMI, and cells were centrifuged for 3 min at 300× *g* and resuspended in 5% FBS in 1× PBS. A total of 10% of cells were fixed with 4% paraformaldehyde for 5 min for replicate analysis with flow cytometry. The remaining cells were filtered with 40 µm filters, centrifuged for 3 min at 300× *g*, and resuspended in 5% FBS in 1× PBS. Cells were sorted using a BD FACSAria II sorter (BD Biosciences, San Jose, CA, USA) with a 70 μm nozzle, serially gating for FSC-A/SSC-A, FSC-H/FSC-W, and GFP+/AmCyan- populations. Sorted cells were collected in 5% FBS in 1× PBS for analysis.

### 4.12. Apoptosis Antibody Array

Monoculture differentiation day 35 hiPSC-CMs were assessed using the 43-target Human Apoptosis Antibody Array (Abcam, ab134001, Cambridge, UK) according to manufacturer’s protocol. Briefly, hiPSC-CMs were lysed with provided lysis buffer and blocked membranes were incubated with 200 μg lysate. Membranes were serially incubated with biotin-conjugated anti-cytokines and streptavidin-HRP. Chemiluminescence images were obtained using the ChemiDoc imaging system as above. Volumetric intensities were analyzed using Bio Rad Image Lab software as above.

### 4.13. RT-qPCR

RNA was isolated from hiPSC-CMs with PureLink RNA minipreps (Invitrogen, 12183018A, Waltham, MA, USA) with PureLink DNase treatment (Invitrogen, 12185010, Waltham, MA, USA) per manufacturer’s protocol. cDNA was generated via reverse transcription using a SensiFast cDNA Synthesis Kit (Thomas Scientific, C755H65, Swedesboro, NJ, USA) per manufacturer’s protocol. qPCR was performed with 10 μL reactions in technical triplicate and 40 cycles with SYBR Select Master Mix (Invitrogen, 4472919, Waltham, MA, USA), 5 ng cDNA per reaction, and 100 nM of forward and reverse primer (Table 3) on an ABI 7900HT Real-Time PCR machine (Fisher, 4329001, Hampton, NH, USA).

### 4.14. Statistics

Statistical comparisons of cell size, forward scatter area, twitch force, cell count, protein expression, and gene expression were performed using one-tailed Student’s *t*-tests with unequal variances and significance criteria *p* < 0.05.

## Figures and Tables

**Figure 1 ijms-24-04909-f001:**
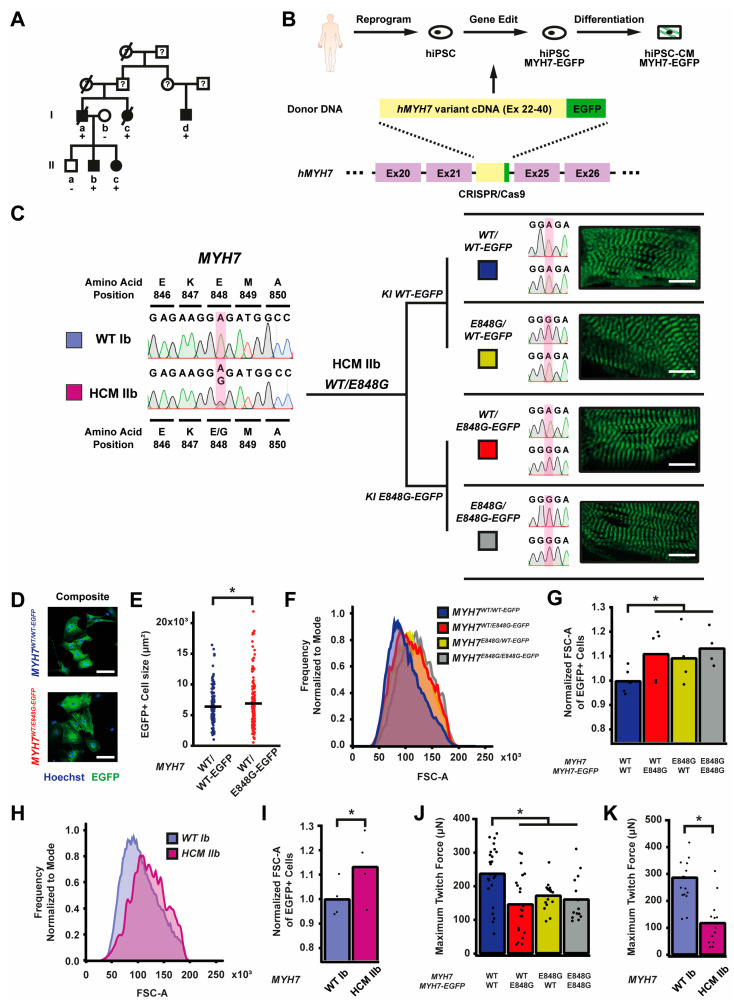
*MYH7* E848G increases cardiomyocyte size and reduces engineered heart tissue contractility in human stem cell model. (**A**) Family pedigree for *MYH7* E848G. (+), WT/E848G; (-), WT/WT; black, HCM; white, no HCM. (**B**) Schematic of CRISPR/Cas9 gene editing strategy to generate isogenic hiPSC-CMs expressing βMHC-EGFP fusion protein under control of endogenous *MYH7* locus. (**C**) (L) Sanger sequencing chromatograms of *MYH7* Exon 22 for patient-derived hiPSCs. (R) Schematic of relationship between isogenic hiPSC-CM lines, with Sanger sequencing chromatograms of *MYH7* Exon 22 and representative confocal microscopy images of sarcomeric striations at differentiation day 35. Scale bar 10 μm. (**D**) Representative confocal microscopy images for Hoechst-stained *MYH7^WT/WT-EGFP^* and *MYH7^WT/E848G-EGFP^* hiPSC-CMs, differentiation day 33. Scale bar 100 µm. (**E**) Two-dimensional projected area of EGFP^+^ *MYH7^WT/WT-EGFP^* and *MYH7^WT/E848G-EGFP^* hiPSC-CMs, differentiation day 33, from confocal images. Mean and cell replicates shown, n = 150 cells. (**F**) Representative histogram of forward scatter area (FSC-A) as measured with flow cytometry of isogenic hiPSC-CMs, differentiation day 40. n = 10,000 cells. (**G**) Normalized FSC-A as measured with flow cytometry of EGFP^+^ isogenic hiPSC-CMs, differentiation day 42. Mean and biological replicates shown. (**H**) Representative histogram of forward scatter area (FSC-A) as measured with flow cytometry of patient-derived hiPSC-CMs, differentiation day 40. n = 10,000 cells (**I**) Normalized FSC-A as measured with flow cytometry of cardiac troponin (cTnT)^+^ patient-derived hiPSC-CMs, differentiation day 42. Mean and biological replicates shown. (**J**) Maximum twitch force of engineered heart tissues (EHTs) at cast week 3 for isogenic hiPSC-CMs. Mean and tissue replicates shown. (**K**) Maximum twitch force of EHTs at cast week 3 for patient-derived hiPSC-CMs. Mean and tissue replicates shown. * in (**E**,**G**,**I**–**K**) indicates *p* < 0.05 significance calculated with Student’s *t*-test.

**Figure 2 ijms-24-04909-f002:**
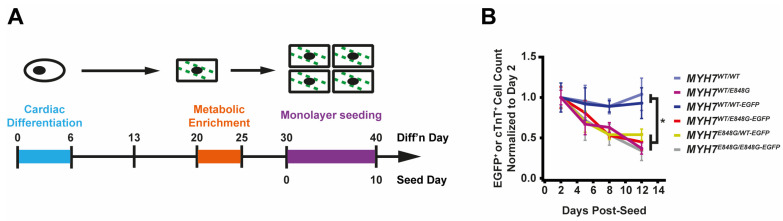
*MYH7* E848G reduces hiPSC-CM survival in monolayer and EHT culture. (**A**) Schematic of iPSC-CM differentiation and seeding protocol for monolayer culture. (**B**) Total EGFP+ cell count for isogenic hiPSC-CMs and cTnT^+^ cell count for patient-derived hiPSC-CMs in DMEM-based monolayer culture, normalized to day 2 post-seed. Mean ± SEM, n = 4–6 biological replicates. * indicates *p* < 0.05 significance calculated with Student’s *t*-test for relative survival values at day 12 post-seed. (**C**) Representative confocal microscopy images for Hoechst- and TUNEL-stained *MYH7*^WT/WT-EGFP^ and *MYH7*^WT/E848G-EGFP^ hiPSC-CMs, differentiation day 33. Positive TUNEL signal indicated with arrows. Scale bar 50 μm. (**D**) Quantification of percentage of TUNEL^+^ nuclei in EGFP^+^ hiPSC-CMs from (**C**), n = 2 biological replicates shown, n = 150 cells per replicate. (**E**) Schematic for casting protocol for engineered heart tissues (EHTs) and papain-based digestion for FACS-based analysis. (**F**) Representative histogram of EGFP intensity as measured with flow cytometry in *MYH7*^WT/WT^, *MYH7*^WT/WT-EGFP^, and *MYH7*^WT/E848G-EGFP^ EHTs, cast week 1. n = 10,000 cells. (**G**) Total EGFP^+^ cells as percentage of initial EHT cast input for *MYH7*^WT/WT-EGFP^ and *MYH7*^WT/E848G-EGFP^ EHTs, cast week 1 and 3, as measured with FACS. Mean and cast replicates shown, 3–6 tissues per cast. (**H**) Normalized FSC-A of EGFP^+^ cells in *MYH7*^WT/WT-EGFP^ and *MYH7*^WT/E848G-EGFP^ EHTs, cast week 1 and 3, as measured with fluorescence-activated cell sorting (FACS). Mean and cast replicates shown, 3–6 tissues per cast. * in (**D**,**G**,**H**) indicates *p* < 0.05 significance calculated with Student’s *t*-test.

**Figure 3 ijms-24-04909-f003:**
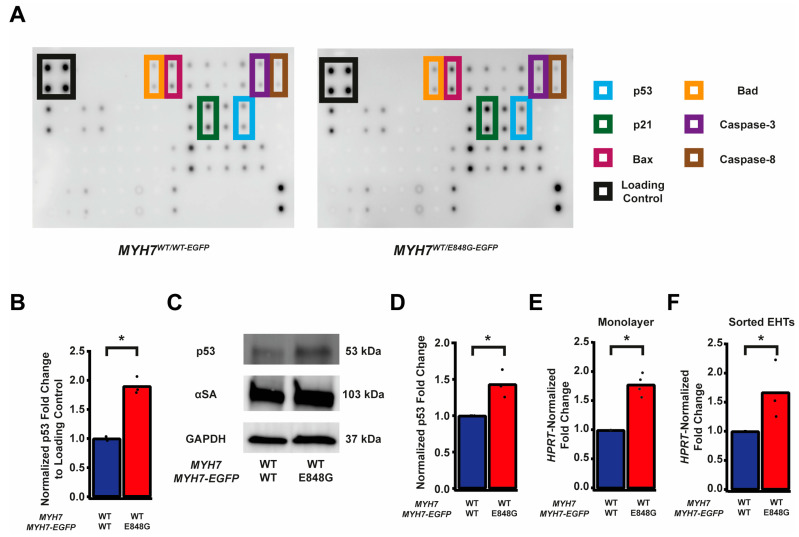
*MYH7* E848G induces p53-associated intrinsic apoptosis in hiPSC-CMs cultured in a monolayer and EHT. (**A**) Human apoptosis antibody array for *MYH7*^WT/WT-EGFP^ and *MYH7*^WT/E848G-EGFP^ hiPSC-CMs, differentiation day 35, with select targets highlighted. (**B**) Quantification of protein expression from (**A**) normalized to loading control and *MYH7*^WT/WT-EGFP^. Mean and biological replicates shown, n = 3 technical replicates. (**C**) Representative Western blot for p53 and α-sarcomeric actinin (αSA) protein expression in *MYH7*^WT/WT-EGFP^ and *MYH7*^WT/E848G-EGFP^ hiPSC-CMs, differentiation day 40. (**D**) Quantification of p53 protein expression from (**C**) normalized to α-sarcomeric actinin and *MYH7*^WT/WT-EGFP^. Mean and biological replicates shown. (**E**) *TP53* mRNA expression for *MYH7*^WT/WT-EGFP^ and *MYH7*^WT/E848G-EGFP^ hiPSC-CMs normalized to *HPRT* and *MYH7*^WT/WT-EGFP^, differentiation day 40, as measured with RT-qPCR. Mean and biological replicates shown. (**F**) *TP53* mRNA expression for *MYH7*^WT/WT-EGFP^ and *MYH7^WT/E848G-EGFP^* EGFP^+^ hiPSC-CMs sorted from cast week 1 EHTs, normalized to *HPRT* and *MYH7^WT/WT-EGFP^*, as measured with RT-qPCR. Mean and biological replicates shown. * in (**B**,**D**–**F**) indicates *p* < 0.05 significance calculated with Student’s *t*-test.

**Figure 4 ijms-24-04909-f004:**
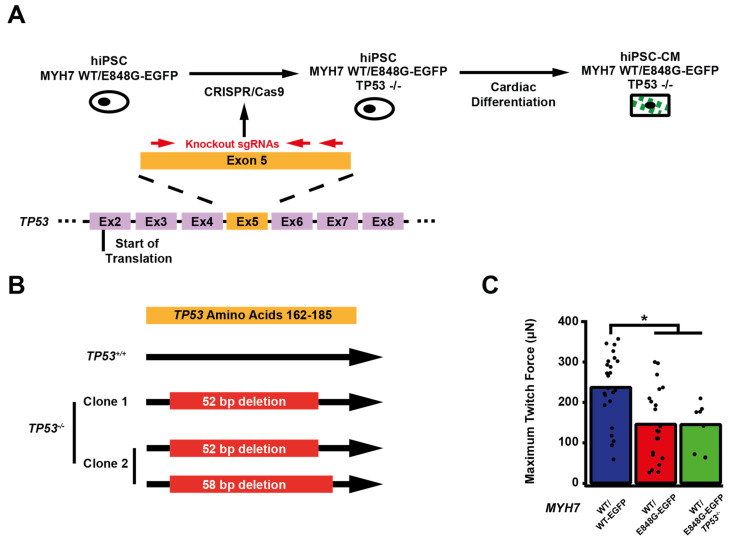
TP53 ablation does not rescue contractile function, hypertrophy, or cardiomyocyte survival in *MYH7* E848G-expressing hiPSC-CMs. (**A**) Schematic of CRISPR/Cas9 gene editing strategy to generate *MYH7*^WT/E848G-EGFP^ *TP53^−/−^* hiPSC-CMs. (**B**) Schematic of deletions in *MYH7*^WT/E848G-EGFP^ *TP53^−/−^* clones. (**C**) Maximum twitch force of EHTs at cast week 3 for isogenic hiPSC-CMs. Mean and tissue replicates shown. (**D**) Representative histogram of EGFP intensity as measured with flow cytometry in *MYH7*^WT/WT^, *MYH7*^WT/WT-EGFP^, *MYH7*^WT/E848G-EGFP^, and *MYH7*^WT/E848G-EGFP^ *TP53^−/−^* EHTs, cast week 1. n = 10,000 cells. (**E**) Total EGFP^+^ cells as percentage of initial EHT cast input for *MYH7*^WT/WT-EGFP^, *MYH7*^WT/E848G-EGFP^, and *MYH7*^WT/E848G-EGFP^ *TP53^−/−^* EHTs, cast week 1 and 3, as measured by FACS. Mean and cast replicates shown, 3–6 tissues per cast. (**F**) Normalized FSC-A of EGFP^+^ cells in *MYH7*^WT/WT-EGFP^, *MYH7*^WT/E848G-EGFP^, and *MYH7*^WT/E848G-EGFP^ *TP53^−/−^* EHTs, cast week 1 and 3, as measured with FACS. Mean and cast replicates shown, 3–6 tissues per cast. * in (**C**,**E**,**F**) indicates *p* < 0.05 significance calculated with Student’s *t*-test.

**Table 1 ijms-24-04909-t001:** Apoptosis antibody array results. Mean fold change of *MYH7*^WT/E848G-EGFP^ relative to *MYH7*^WT/WT-EGFP^, n = 3 biological and n = 2 technical replicates per target. * indicates *p* < 0.05 significance calculated with Student’s *t*-test.

Target	Mean	SEM	*p*-Value	Significance
Bad	1.6517	0.1442	0.0027	*
Bax	1.7381	0.2500	0.0182	*
Bcl-2	1.8773	0.2640	0.0085	*
Bcl-w	1.8071	0.2904	0.0160	*
BID	1.6310	0.1420	0.0027	*
BIM	1.5624	0.1580	0.0064	*
Caspase 3	1.8363	0.2449	0.0056	*
Caspase 8	1.5411	0.0642	0.0002	*
CD40	1.3168	0.2759	0.2892	n.s.
CD40L	2.0300	0.8439	0.2153	n.s.
cIAP-2	1.9278	0.6928	0.1650	n.s.
cytoC	Undetected	Undetected	Undetected	n.s.
DR6	Undetected	Undetected	Undetected	n.s.
Fas	Undetected	Undetected	Undetected	n.s.
FasL	1.7717	0.1051	0.0002	*
HSP27	1.7563	0.2584	0.0122	*
HSP60	1.8974	0.0777	0.0000	*
HSP70	2.3238	0.7780	0.0614	n.s.
HTRA	Undetected	Undetected	Undetected	n.s.
IGF-1sR	Undetected	Undetected	Undetected	n.s.
IGFBP-1	Undetected	Undetected	Undetected	n.s.
IGFBP-2	Undetected	Undetected	Undetected	n.s.
IGFBP-3	Undetected	Undetected	Undetected	n.s.
IGFBP-4	Undetected	Undetected	Undetected	n.s.
IGFBP-5	Undetected	Undetected	Undetected	n.s.
IGFBP-6	Undetected	Undetected	Undetected	n.s.
IGF-I	Undetected	Undetected	Undetected	n.s.
IGF-II	1.7037	0.1599	0.0022	*
Livin	1.9295	0.2734	0.0056	*
p21	1.8962	0.1998	0.0024	*
p27	1.9002	0.0846	0.0000	*
p53	1.7654	0.1552	0.0012	*
SMAC	Undetected	Undetected	Undetected	n.s.
sTNF-R1	Undetected	Undetected	Undetected	n.s.
sTNF-R2	1.7433	0.1947	0.0320	*
Survivin	1.2634	0.2632	0.3647	n.s.
TNFα	1.7563	0.4123	0.0807	n.s.
TNFβ	Undetected	Undetected	Undetected	n.s.
TRAILR-1	Undetected	Undetected	Undetected	n.s.
TRAILR-2	2.0103	0.3491	0.0216	*
TRAILR-3	Undetected	Undetected	Undetected	n.s.
TRAILR-4	1.7835	0.2270	0.0066	*
XIAP	1.6517	0.1442	0.0027	*

**Table 2 ijms-24-04909-t002:** sgRNA sequences for CRISPR/Cas9 editing.

Guide Target	Sequence
*MYH7*	UUCAUAUGAGCCCCUCCUGC
*MYH7*	GCCUUUGACACAAGAUUUAG
*TP53*	CGCUAUCUGAGCAGCGCUCA
*TP53*	GUGCUGUGACUGCUUGUAGA
*TP53*	CAACAAGAUGUUUUGCCAAC

**Table 3 ijms-24-04909-t003:** Primer sequences for Sanger sequencing, PCR, and RT-qPCR.

Primer Target	Condition	Forward (5′–3′)	Reverse (5′–3′)	Note
*MYH7*	Sequencing	AGACTCCCTGCTGGTAATCCAGTG	N/A	
*MYH7*	PCR	ATCCCTGAGGGACAGTTCATTG	GGGTTGTGGGAAGTGAAGGC	Amplifies native allele
*MYH7*	PCR	ATCCCTGAGGGACAGTTCATTG	GGTTGTCTTGTTCCGCCTG	Amplifies knockin allele
*TP53*	PCR	CGCCAACTCTCTCTAGCTCG	GCACCACCACACTATGTCGA	
*HPRT*	RT-qPCR	TGACACTGGCAAAACAATGCA	GGTCCTTTTCACCAGCAAGCT	
*TP53*	RT-qPCR	CAGCACATGACGGAGGTTGT	TCATCCAAATACTCCACACGC	

**Table 4 ijms-24-04909-t004:** Western blot antibodies.

Target	Species	Dilution	Vendor	Catalog Number
βMHC	Mouse	1:500	Developmental Studies Hybridoma Bank, Iowa City, IA, USA	A4.951
GFP	Goat	1:250	Novus Bio, Englewood, CO, USA	100-1770
α-Sarcomeric Actinin	Rabbit	1:1000	Abcam, Cambridge, UK	AB68167
GAPDH	Mouse	1:2000	Santa Cruz Biotechnology, Dallas, TX, USA	SC32233
p53	Mouse	1:500	Santa Cruz Biotechnology, Dallas, TX, USA	SC126
anti-Mouse HRP	Goat	1:2000	Bio Rad, Hercules, CA, USA	1705047
anti-Rabbit HRP	Goat	1:2000	Bio Rad, Hercules, CA, USA	1705046

## Data Availability

The data presented in this study are openly available at Zenodo, https://doi.org/10.5281/zenodo.7563492.

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
