# Peer review of "Cardiomyocyte Apoptosis Is Associated with Contractile Dysfunction in Stem Cell Model of MYH7 E848G Hypertrophic Cardiomyopathy"

_ijms, 2023, doi:10.3390/ijms24054909_

Round 1

Reviewer 1 Report

Cardiac hypertrophy is a common genetic disorder featured by increased left ventricle mass and impaired contractile function. Mutations in several myofilament proteins including MYH7 can cause hypertrophic cardiomyopathy (HCM). In this study, Loiben et al. investigated the mechanism underlying MYH7 mutation-associated HCM by integrating Crisp-cas9-based gene-editing and patient-derived iPSC. Heterozygous MYH7E848G/+ in iPSC increased cardiomyocyte size and disturbed the contractility of engineered heart tissue.  These observations recapitulate phenotypic changes in MYH7E848G mutated patients. Further, MYH7E848G/+ cardiomyocytes exhibit increased pro-apoptotic p53 activity and apoptosis fraction. Interestingly, deleting TP53 did not alter MYH7E848G/+ cardiomyocyte apoptosis and contractility. While this study may provide a valuable tool as a patient-derived iPSC cell line for MYH7-related HCM treatment screening, more evidence needs to be provided to support the proposed conclusion that cardiomyocyte apoptosis is a key player in MYH7E848G/+ HCM.

Major concerns:

1. The authors tend to believe apoptosis can be responsible for MYH7E848G/+ -related hypertrophy and systolic dysfunction. In that case, apoptosis inhibitors such as caspase-3 inhibitor (PMID: 32610140) should be applied to demonstrate that inhibiting cardiomyocyte apoptosis can rescue or improve the cardiomyocyte size and contractility dysregulation since the apoptotic signal pathway ultimately leads to activation of the executioner caspase 3.

2. Apoptosis antibody array results show the tendency of increased intrinsic apoptotic proteins including Bad and Bax in MYH7E848G/+ cells. With n =2 in this experiment, it is difficult to draw any statistical conclusion here.  Experimentation to validate such observation is highly recommended.

3. Despite the high efficiency of CRISPR/Cas9 system-mediated gene deleting, the ablation of TP53 should be confirmed by WB or other approaches. Particularly, the result that TP53 deletion did not change the iPSC phenotype indicates the deletion may be not very efficient.

Minor concerns:

1. Please highlight the positive TUNEL signal in both the “TUNEL” and “Composite” panels. 

2. In panel B of Fig 2, it is not clear whether the authors were comparing the survival curve, or the last time point survival ratio, which requires different statistical analysis. So, please clarify it.

3. Scale bar is included in most of the images except several images in Figure 1.

4. Please add the molecular weight to the blot images.

Author Response

We thank you for your thoughtful and considerate comments regarding our recent submission titled “Cardiomyocyte apoptosis contributes to contractile dysfunction in stem cell model of MYH7 E848G hypertrophic cardiomyopathy”. 

With our revised submission and enclosed comments, we would like to respond to the reviewers’ concerns.

Cardiac hypertrophy is a common genetic disorder featured by increased left ventricle mass and impaired contractile function. Mutations in several myofilament proteins including MYH7 can cause hypertrophic cardiomyopathy (HCM). In this study, Loiben et al. investigated the mechanism underlying MYH7 mutation-associated HCM by integrating Crisp-cas9-based gene-editing and patient-derived iPSC. Heterozygous MYH7E848G/+ in iPSC increased cardiomyocyte size and disturbed the contractility of engineered heart tissue.  These observations recapitulate phenotypic changes in MYH7E848G mutated patients. Further, MYH7E848G/+ cardiomyocytes exhibit increased pro-apoptotic p53 activity and apoptosis fraction. Interestingly, deleting TP53 did not alter MYH7E848G/+ cardiomyocyte apoptosis and contractility. While this study may provide a valuable tool as a patient-derived iPSC cell line for MYH7-related HCM treatment screening, more evidence needs to be provided to support the proposed conclusion that cardiomyocyte apoptosis is a key player in MYH7E848G/+ HCM.

The authors tend to believe apoptosis can be responsible for MYH7E848G/+ -related hypertrophy and systolic dysfunction. In that case, apoptosis inhibitors such as caspase-3 inhibitor (PMID: 32610140) should be applied to demonstrate that inhibiting cardiomyocyte apoptosis can rescue or improve the cardiomyocyte size and contractility dysregulation since the apoptotic signal pathway ultimately leads to activation of the executioner caspase 3.

While our data demonstrates an association of apoptosis in the MYH7E848G/+ hiPSC-CMs, we acknowledge that it does not confirm that apoptosis contributes to the HCM phenotype, but instead it suggests there is an association of apoptosis and the E848G allele. The TUNEL stain of the hiPSC-CM monolayer and activated Caspase-3 and Caspase-8 on the apoptosis antibody array demonstrate an association of apoptosis with the E848G allele (Figure 2c-d, Table 1), while the loss of cells in a monolayer (Figure 2b) and in tissues (Figure 2g) further suggests cell death is associated with this pathogenic variant.  We believe the proposed experiment to test for the effect of caspase inhibition has merit to confirm that apoptosis contributes to the phenotype, such as using a pan-caspase inhibitor, Z-VAD-FMK. However, such an experiment will require more than the allotted time for response. There are also caveats applicable to the use of caspase inhibitors in this context. It is plausible that blocking the execution step of apoptosis will yield a larger population of dysfunctional cardiomyocytes instead of restoring function to the apoptotically activated cells. We have modified the title of the manuscript and the abstract (line 28) to clearly describe this new association of cardiomyocyte apoptosis with the pathogenic MYH7 E848G variant.

Apoptosis antibody array results show the tendency of increased intrinsic apoptotic proteins including Bad and Bax in MYH7E848G/+ cells. With n =2 in this experiment, it is difficult to draw any statistical conclusion here.  Experimentation to validate such observation is highly recommended.

We appreciate the concern regarding the statistical significance of our apoptosis antibody array results. We have included an additional biological replicate of the apoptosis antibody array for MYH7WT/E848G-EGFP and MYH7WT/WT-EGFP to confirm these findings (Table 1), and included p-values generated by t-tests. With n = 3, we observed significant upregulation in protein expression of Bad, Bax, Caspase-3, Caspase-8, p21, and p53, among other signals (Fig. 3a-b, Fig. S3a-e). These targets have been highlighted in revised Figure 3a and Figure S3a-e. With this additional replicate, we are confident in the overall trend of increased apoptotic protein levels in MYH7E848G/+ cells.

Despite the high efficiency of CRISPR/Cas9 system-mediated gene deleting, the ablation of TP53 should be confirmed by WB or other approaches. Particularly, the result that TP53 deletion did not change the iPSC phenotype indicates the deletion may be not very efficient.

We understand the skepticism associated with a gene editing strategy that does not result in a change in phenotype. We generated two distinct single clonal lines with the deletion of 52 and/or 58 base pairs in each TP53 allele as confirmed by Sanger sequencing. Clone #1 has homozygous 52 bp deletions in TP53, while clone #2 has a 52 bp deletion in one allele and a 58 bp deletion in the second allele. This results in the removal of at least 17 amino acids and a corresponding frame shift in Exon 5 of 11; Exons 5-8 correspond with the DNA-binding domain of p53 (PMID: 12826609). Thus, we are confident in the ablation of TP53 in our lines. Additional confirmatory Western blots will require more than the allotted time for response because we will need to generate additional protein lysates.

Please highlight the positive TUNEL signal in both the “TUNEL” and “Composite” panels.

We have added identifying arrows to the corresponding panels in Figure 2c to aid in interpretation of the results as requested.

In panel B of Fig 2, it is not clear whether the authors were comparing the survival curve, or the last time point survival ratio, which requires different statistical analysis. So, please clarify it.

We appreciate the reviewer highlighting the ambiguity of the markers of significance in Figure 2b. As such, we have added the following text to the figure caption to reflect the correspondence of the statistical testing to the last time point survival ratio.

Line 161: “* indicates p < 0.05 significance calculated by Student’s t-test for relative survival values at day 12 post-seed.

Scale bar is included in most of the images except several images in Figure 1.

The images in Figure 1C and 1D share the same scale bar as their respective first images in their subfigures. For clarity, we have added scale bars to the other images as well.

Please add the molecular weight to the blot images.

The publication example Western blots have been updated with the molecular weight indicators as requested.

Reviewer 2 Report

By targeting endogenous MYH7 locus in the hiPSC lines previously derived from an HCM patient, the authors in this manuscript successfully generated four isogenic hiPSC lines with one MYH7 allele EGFP-tagged, with or without the MYH7E848G/+ variant. The authors demonstrated that MYH7WT/E848G-EGFP hiPSC-CMs increase cell size and reduce  contractility measured by maximum twitch forces of engineered heart tissue, recapitulating the key measures of hypertrophy and hypocontractility present in vivo. The authors also observed that MYH7E848G/+ cardiomyocytes exhibit reduced survival due to apoptosis in a P53-independent manner. Although the authors did not figure out the molecular mechanism underlying MYH7E848G/+ HCM, they established isogenic cell line models that are beneficial for future study to understand the mechanisms of HCM.

1.Fig1a shows Id is the new patient, but manuscript line 79 indicates patient IId.

2.FigS1a, I don’t find pJet-MYH7-EGFP-PGK-PuroR plasmid information in Supp.File1.

3.Fig1b, schematic of hMYH7 variant cDNA indicates exon 21-40, but Fig S1a indicates exon 22-40, it is not consistent.

4.Because the four established cell lines are analyzed in Fig1, can βMHC expression of the four lines  be tested in FigS1C to confirm that no preferential expression of one allele over the other in all four lines?

5. α-sarcomeric actinin protein is absent in the western blot result in Fig3C.

6. Apoptosis is probably the outcome of some cellular events related to MYH7E848G mutation, does the MYH7 E848G mutation affect cardiac differentiation or βMHC stability?

Round 2

Reviewer 1 Report

The manus has been improved, and the authors have addressed my concerns appropriately.